# Distinctive Kaposi Sarcoma-Associated Herpesvirus Serological Profile during Acute *Plasmodium falciparum* Malaria Episodes

**DOI:** 10.3390/ijms24076711

**Published:** 2023-04-04

**Authors:** Peter O. Oluoch, Catherine S. Forconi, Cliff I. Oduor, Dominic A. Ritacco, Hoseah M. Akala, Jeffrey A. Bailey, Jonathan J. Juliano, John M. Ong’echa, Christian Münz, Ann M. Moormann

**Affiliations:** 1Division of Infectious Diseases and Immunology, Department of Medicine, Chan Medical School, University of Massachusetts, Worcester, MA 01605, USA; 2Center for Global Health Research, Kenya Medical Research Institute, Kisumu 40100, Kenya; 3Department of Pathology and Laboratory Medicine, Warren Alpert Medical School, Brown University, Providence, RI 02903, USA; 4Division of Infectious Diseases, Department of Medicine, School of Medicine, University of North Carolina at Chapel Hill, Chapel Hill, NC 27599, USA; 5Department of Epidemiology, Gillings School of Global Public Health, University of North Carolina, Chapel Hill, NC 27599, USA; 6Viral Immunobiology, Institute of Experimental Immunology, University of Zurich, CH-8057 Zurich, Switzerland

**Keywords:** Kaposi sarcoma-associated herpesvirus, Epstein–Barr virus, acute malaria infection, endemic Kaposi sarcoma, sub-Saharan Africa, lytic replication, Kenya

## Abstract

The seroprevalence of Kaposi sarcoma-associated herpesvirus (KSHV) and the incidence of endemic Kaposi sarcoma (KS) overlap with regions of malaria endemicity in sub-Saharan Africa. Multiple studies have shown an increased risk of KSHV seroconversion in children from high malaria compared to low malaria regions; however, the impact of acute episodes of *Plasmodium falciparum* (*P. falciparum*) malaria on KSHV’s biphasic life cycle and lytic reactivation has not been determined. Here, we examined KSHV serological profiles and viral loads in 134 children with acute malaria and 221 healthy children from high malaria regions in Kisumu, as well as 77 healthy children from low malaria regions in Nandi. We assayed KSHV, Epstein–Barr virus (EBV), and *P. falciparum* malaria antibody responses in these three by multiplexed Luminex assay. We confirmed that KSHV seroprevalence was significantly associated with malaria endemicity (OR = 1.95, 1.18–3.24 95% CI, *p* = 0.01) with 71–77% seropositivity in high-malaria (Kisumu) compared to 28% in low-malaria (Nandi) regions. Furthermore, KSHV serological profiles during acute malaria episodes were distinct from age-matched non-malaria-infected children from the same region. Paired IgG levels also varied after malaria treatment, with significantly higher anti-ORF59 at day 0 but elevated ORF38, ORF73, and K8.1 at day 3. Acute malaria episodes is characterized by perturbation of KSHV latency in seropositive children, providing further evidence that malaria endemicity contributes to the observed increase in endemic KS incidence in sub-Saharan Africa.

## 1. Introduction

Kaposi sarcoma-associated herpesvirus (KSHV) is an oncovirus that causes Kaposi sarcoma (KS) and other diseases, including primary effusion lymphoma and a form of multicentric Castleman disease (KSHV-MCD) [1]. The seroprevalence of KSHV is unaccountably high in malaria-endemic areas of sub-Saharan Africa [2,3]. Studies from these regions have reported a significant correlation between KSHV- and malaria-specific antibodies in both children and adults [4,5,6,7,8]. Prior to the HIV/AIDS epidemic, endemic Kaposi sarcoma (KS) incidence was estimated at 6 per 100,000 person-year in East Africa, including Uganda, Tanzania, and Zaire (now the Democratic Republic of Congo) compared to an incidence of 0.5–1.5 per 100,000 person-year in north and south African countries [9]. Though multiple theories have been proposed to explain the strong geographic overlap in malaria endemicity, KSHV seroprevalence, and endemic KS incidence in sub-Saharan Africa, including quinine-linked “oncoweed” [10] and “oncodrug” hypotheses [11], they are inconsistent with stable KSHV seroprevalence in malaria-endemic regions years after the replacement of quinine as an anti-malarial. While previous studies have implicated *Plasmodium falciparum* (*P. falciparum*) malaria infections with inducing KSHV lytic replication and by extension, the strong geographical clustering of malaria endemicity with KSHV seroprevalence, changes in KSHV serological profiles during acute malaria episodes have not yet been studied.

By comparison, the interactions between *P. falciparum* malaria and another herpesvirus, Epstein–Barr virus (EBV) in the etiology of endemic Burkitt lymphoma (eBL) have been extensively described [12,13,14]. Several studies have shown that persistent replicative blood-stage malaria infections in children are associated with transient activation of EBV from latency to a viremic lytic state, higher steady-state viral loads, and an increased risk of eBL development [14,15,16]. Cognate surface receptors in EBV-infected memory B-cells interact with the cysteine-rich inter-domain region 1 alpha (CIDR1ɑ) domain, specifically of *P. falciparum* erythrocyte membrane protein 1 (*Pf*EMP1) on parasitized red blood cells (pRBCs). This direct interaction stimulates B cell proliferation and EBV reactivation [15,16,17,18,19]. Though less studied, a similar model involving interactions between CD36-expressing KSHV-infected B cells and the *Pf*EMP1 CIDR1ɑ domain has been implicated in KSHV lytic replication [16]. Conant et al. (2013) propose a model in which such an interaction initiates a transcriptional cascade, culminating in the sequential transcriptional activation of immediate early (IE), delayed early (DE) and structural KSHV lytic cycle genes [16]. While this proposed model provides a possible link between malaria, KSHV lytic replication and endemic KS distribution, the extent of this relationship remains largely unexplored in at-risk populations [5,8,20]. Though acute malaria infection perturbs EBV latency, characterized by the initiation of a lytic transcriptional program and increased viremia, the effects of acute malaria episodes on KSHV lytic replication have yet to be determined. 

In this study, we evaluated KSHV serological patterns indicative of lytic replication in children during an episode of acute malaria and compared them to two separate groups of non-malaria-infected Kenyan children. We demonstrate that acute malaria subjects exhibit a distinct pattern of elevated antibodies against KSHV lytic antigens, indicating that malaria infection induces KSHV reactivation.

## 2. Results

### 2.1. Characteristics of Study Participants

To determine the seroprevalence of KSHV in malaria-endemic areas, we examined acute malaria-infected and healthy subjects from a region with high incidence of malaria (Kisumu) as well as healthy subjects from a region with low incidence of malaria (Nandi). A total of 134 children (median age 4.8, IQR; 3.4–6.9) experiencing an episode of uncomplicated acute *P. falciparum* malaria from Kisumu (A-KSM) were compared to 221 (median age 5.6, IQR; 4.5–8.7) healthy children from Kisumu (H-KSM), and 77 (median age 5.3, IQR; 3.3–5.5) healthy children from Nandi (H-NDI). As expected, KSHV seroprevalence correlated with levels of malaria transmission, with significantly higher seroprevalence in high-malaria Kisumu children compared to low-malaria Nandi children (*p* < 0.0001 by Pearson’s Chi-squared test; Table 1).

### 2.2. Acute P. falciparum Malaria Is Characterized by Distinct Serological Profiles in KSHV Seropositive Children

We next compared the serological profiles of the three cohorts and found that IgG antibodies against malaria antigens, MSP1 and AMA1, were significantly elevated in KSHV seropositive compared to seronegative children (Figure 1a), a pattern supported by additional multivariate regression analysis of seropositivity determinants when controlling for the confounding effects of participants’ age and gender (Figure 1b). Significantly higher odds of KSHV seropositivity were observed in children from the high-malaria region of Kisumu (H-KSM OR = 1.95, 1.18–3.24, 95% CI, *p* = 0.01 and A-KSM OR = 1.95, 1.19–3.35, 95% CI, *p* = 0.01) and in children with high *P. falciparum* malaria anti-MSP1 IgG antibodies (OR = 1.70, 1.07–2.72 95% CI, *p* = 0.025). 

To determine the effects of acute malaria infections on KSHV serological patterns, we first projected individual participants from the three study populations (independent of KSHV sero-status) into a 2-D principal component analysis (PCA) based on previously validated IgG antibody levels to seven KSHV antigens (Figure 1c). A combination of five out of these seven KSHV antigens more accurately defines seropositivity [8]. The first principal component (Dimension 1) clusters participants into two regions based on patterns of malaria transmission intensity: high (A-KSM and H-KSM) and low (H-NDI). Principal component 2 (PC2) explained 13.5% of the differences, with KSHV lytic antigens ORF59, K5, ORF61, and ORF38 presenting with a distinct pattern in A-KSM at day 0 (before the start of anti-malarial treatment). Overall, both the PCA and multivariate regression analysis confirmed the previously described geographical overlap between malaria incidence and KSHV seroprevalence.

### 2.3. KSHV Lytic Antigens Are Increased in Patients with Acute Malaria

To determine the specific KSHV antigens that are elevated during acute malaria, we compared antibody levels to these antigens in the three cohorts. Consistent with our PCA characterization, we identified distinct antibody signatures for the KSHV antigens ORF59, K8.1, ORF38, and ORF73 across the three study populations. IgG antibodies against ORF73 were significantly higher in healthy subjects compared to A-KSM-D0 subjects. K8.1 IgG antibodies were also slightly elevated in H-KSM while ORF59 IgG was slightly elevated in A-KSM-D0 subjects compared to H-KSM (Figure 2a). In contrast, anti-ORF38 IgGs were significantly elevated in acute malaria patients from Kisumu at day 3 compared to healthy children from the same region (Figure 2a). Delayed early antigens such as K5, ORF61, and ORF65 had comparable IgG antibody levels in the three study populations (Appendix A). As a comparator, we examined EBV, which is known to become reactivated during acute malaria. We found EBV anti-VCA but not EBNA1 IgG levels were significantly higher in A-KSM children at day 0 compared to both H-KSM and H-NDI children (Figure 2b) and showed a modest positive correlation with IgG against KSHV antigens, including K5, ORF38, ORF59, and ORF61 (Appendix A). Altogether, these results demonstrate that acute malaria episodes perturb KSHV latency, leading to elevated IgG antibodies against KSHV lytic antigens.

### 2.4. Pairwise Comparison of Antibody Responses Reveals Distinct IgG and IgM Antibodies to KSHV Antigens during Acute Malaria Episodes

To further assess the effects of acute malaria episodes on KSHV latency, we performed a pairwise comparison of anti-KSHV IgG and IgM antibodies in pre- (A-KSM-D0) and post-treatment samples (A-KSM-D3). From the pairwise comparisons, IgG antibodies against ORF38 (*p* < 0.0001), K8.1 (*p* < 0.0001), and ORF73 (*p* = 0.0002) antigens were significantly higher at day 3 compared to day 0 (Wilcoxon matched-pairs test, Figure 3a). Consistent with this IgG pattern, anti-KSHV IgM were significantly elevated in A-KSM at day 3 compared to paired A-KSM samples at day 0 (Figure 3b). To confirm if the serological pattern is indicative of ongoing lytic reactivation, we assayed both KSHV and EBV loads at both recruitment and last day of malaria treatment. In the acute malaria-infected cohort, KSHV load was undetectable both in whole cell pellets and cell-free plasma using droplet digital PCR (ddPCR). This result was not due to an inability to detect KSHV by the assay, as KSHV^+^ controls included in each plate had a consistent range of viral copies/µL of DNA (Appendix A). In contrast to KSHV, the EBV load was significantly elevated in acute malaria relative to healthy Kisumu children (Figure 3c), independent of KSHV seropositivity (Appendix A, left panel), which was consistent with previous studies from the same region [21,22]. Though malaria parasite load at recruitment was independent of KSHV seropositivity (Appendix A, right panel), we observed a modest relationship between parasitemia and IgM antibodies against lytic KSHV antigens. At day 0, IgM antibodies against K5, ORF61, and ORF38 antigens are therefore positively correlated with parasite density (Figure 3d).

## 3. Discussion

In this study, we report the first detailed comparison of a panel of KSHV antibodies in children with acute *P. falciparum* malaria compared to non-malaria-infected healthy children from high (Kisumu) and low (Nandi) malaria transmission regions in Kenya. Our results show a distinct KSHV serological profile during an episode of malaria, reminiscent of KSHV DNA replication, followed by the release of structural virion proteins as well as the KSHV latency genes during virion-induced cell destruction. Our observation is consistent with a recently proposed model of molecular interaction between pRBCs and KSHV-infected memory B-cells [16]. Using paired analysis, we show that IgG responses to the virion proteins ORF38 (tegument) and K8.1, as well as latency-associated nuclear antigen (LANA encoded by ORF73 and released during host cell destruction), were significantly elevated by the third day as acute malaria episode resolved during the 3-day course of anti-malarial treatment. Though KSHV DNA was undetected in peripheral blood, the serological profile points to possible lytic reactivation during acute malaria episodes. This is consistent with studies that have reported a higher risk of KSHV seroconversion with an increasing malaria annualized rate [6].

During acute malaria episodes, previous studies have shown that EBV-specific T cells fail to control EBV-infected B cells, leading to abnormal proliferation, initiation of lytic reactivation, and elevated viral loads [15,23,24]. In our study, A-KSM children presented with significantly higher EBV loads and stronger anti-VCA IgG antibody responses, supporting the existing evidence of the effects of malaria episodes on EBV kinetics. Though studies on how malaria episodes impact KSHV control are lacking, our results show that acute malaria episodes similarly disrupt KSHV latency. Even in the absence of KSHV load data, the variable serological patterns for K8.1 and ORF38 in A-KSM relative to H-KSM are consistent with perturbation of latency. Significantly higher IgG antibody levels against K8.1 in H-KSM vs. A-KSM-D0 but not A-KSM-D3 could also reflect different stages of lytic replication over the course of a malaria infection undergoing treatment. Though elevated levels of K8.1 in healthy vs. acute malaria children is inconsistent with lytic replication patterns, this could be reflective of high reactivity of K8.1 peptides relative to other KSHV antigens [25,26,27]. How this pattern would progress for undiagnosed and, thus, untreated malaria infections remain unknown. However, the same pattern is also evident in IgG antibodies against ORF38; comparable in A-KSM-D0 vs. H-KSM but significantly higher in A-KSM-D3 vs. A-KSM-D0 and H-KSM. Both K8.1 and ORF38 genes are expressed 48–72 h post-initiation of replication [25,26,27]. The increased levels of KSHV-specific IgM antibodies at day 3 in paired samples are also consistent with atypical memory B-cell responses previously associated with persistent exposure to intense malaria transmission [28]. This can also be skewed by polyclonal activation of B cells during acute malaria episodes [29], evident from the positive correlation between anti-K5, ORF61, and ORF38 IgG antibodies and malaria parasite density. Thus, additional studies designed to characterize temporal KSHV transcriptomics in peripheral blood mononuclear cells (PBMCs) are needed to comprehensively define the effects of acute malaria episodes on KSHV lytic reactivation. Such studies may overcome the key limitation here; the failure to detect KSHV load in longitudinal whole blood and plasma samples within our study population. As different genes are amplified for KSHV detection, including ORF26 [8], ORF73 [30] and K6 [31], it is possible that the sensitivity of ORF26 (a late lytic protein) is low and the other viral genes could be explored to increase the sensitivity of KSHV detection. Alternatively, the design of this study and use of DBS-extracted DNA for viral load quantification may have limited the sensitivity of KSHV DNA detection. Nevertheless, malaria seems to drive both EBV and KSHV lytic replication, and future studies are needed to characterize if the reactivated cellular reservoirs of these two viruses overlap in pediatric and adult infections, as would be suggested by recent studies on EBV co-infection supporting KSHV persistence [32,33,34,35,36,37] 

In conclusion, this study provides evidence that acute malaria infections initiate a transition through KSHV lytic replication stages characterized by a serological pattern reflective of the expression of early intermediate, delayed early, and tegument lytic KSHV proteins. Thus, it also provides further evidence that malaria seems to be reactivating KSHV replication, possibly thereby contributing to the increased incidence of endemic KS in the region. Though the average decline in KS incidence is largely associated with a reduction in HIV/AIDS incidence and the rollout of antiretroviral therapy [38], the impacts of effective malaria control programs within sub-Saharan Africa on this downward trend cannot be ignored.

## 4. Materials and Methods

### 4.1. Study Area, Participants, and Ethical Approval

Children presenting with acute uncomplicated *P. falciparum* malaria (referred to as acute Kisumu; A-KSM) were sampled from a larger multicenter study designed to evaluate the efficacy of artemether-lumefantrine (AL) in Ahero, Kisumu County, Kenya between September 2016 and July 2018. Participants were included in the study if they presented with at least 500 parasites per 200 white blood cells (WBC) by microscopy and admitted to Ahero sub-county hospital clinical trial site in Kisumu County for directly observed therapy consisting of 6 doses over 3 days, following the current Ministry of Health (Kenya) guidelines [39]. Whole blood samples for serology, malaria parasitemia, and viral loads were collected at day 0 (A-KSM-D0) prior to treatment initiation and at day 3 (A-KSM-D3) of treatment with AL. Additional control cohorts of healthy children were recruited from Chulaimbo, Kisumu County (healthy Kisumu; H-KSM) and from Mosoriot, Nandi County (healthy Nandi; H-NDI) between 2014 and 2018. Chulaimbo and Ahero sub-counties within Kisumu County are situated 35 km west and 20 km east of Kisumu city, respectively, and both are characterized by perennial high malaria transmission. Conversely, Mosoriot, in Nandi County, is a region with low and seasonal malaria [40]. For inclusion in this study, all participants were HIV-negative and born of HIV-negative mothers. For the healthy control group, only children with undetectable malaria parasites by both microscopy and qPCR were included. 

### 4.2. Antibody Serology Assay

Using multiplexed bead-based assay, we measured the levels of IgG and IgM antibodies to 7 previously described KSHV antigens [8,41] and antibodies against EBV (EBNA1 and VCA) and malaria (AMA1 and MSP1) antigens in A-KSM-D0, A-KSM-D3, H-KSM, and H-NDI children. Based on previously described protocols [8,42], Bio-plex COOH carboxylated beads were coupled to recombinant KSHV open reading frames (ORFs), EBV, and malaria antigens. IgG and IgM antibodies mean fluorescence intensities (MFI) were obtained from BioPlex 200 Multianalyte Analyzer (Bio-Rad Laboratories, Hercules, CA, USA). Each plate included two positive and negative controls, and the percent relative standard deviation was used to determine inter/intra assay variability. KSHV seropositivity cutoffs were determined based on the receiver operating characteristic (ROC) curve [8] and an analysis of IgG antibody levels in two high and low KSHV exposure risk populations was conducted as described [8]. For A-KSM, KSHV seropositivity was based on day 0 IgG antibody profiles.

### 4.3. Parasite Density and Viral Load Quantification

To determine malaria parasite density and viral load, DNA was extracted from dry blood spots (DBS, with 50µl blood equivalent) using Qiagen QiaAMP DNA extraction kits (Qiagen, Hilden, Germany) in accordance with the manufacturer’s instructions. DNA was suspended in the elution buffer, followed by *P. falciparum* malaria quantification by real-time PCR (qPCR) amplification of the lactate dehydrogenase (*pfldh*) gene [43]. Droplet digital PCR (ddPCR) amplification of *ORF26* and *BALF5* genes was used to quantify KSHV and EBV loads, respectively, relative to the human β-actin gene. Previously published *ORF26* [8], human β-actin, and *BALF5* [44] primers and PCR conditions were used, and the KSHV and EBV load was expressed as viral copies/µg human DNA (hDNA).

### 4.4. Statistical Analysis

All statistical analyses and data visualization were performed using R (version 3.6.3) and GraphPad Prism (v9.0.0). KSHV seropositivity cutoff was estimated using ROC analysis as previously described using MFI values from 11 African and 15 North American control samples [8]. Wilcoxon rank-sum test with Benjamini–Hochberg (BH) correction and Wilcoxon matched-pairs signed rank tests were used for comparison of KSHV, EBV and malaria antibody levels. Rank-based Spearman correlation was used to estimate the relationship between different variables as implemented in the ggscatter R package. Principal component analysis (PCA) was implemented in R package *prcomp* with a scaled option. 

## Figures and Tables

**Figure 1 ijms-24-06711-f001:**
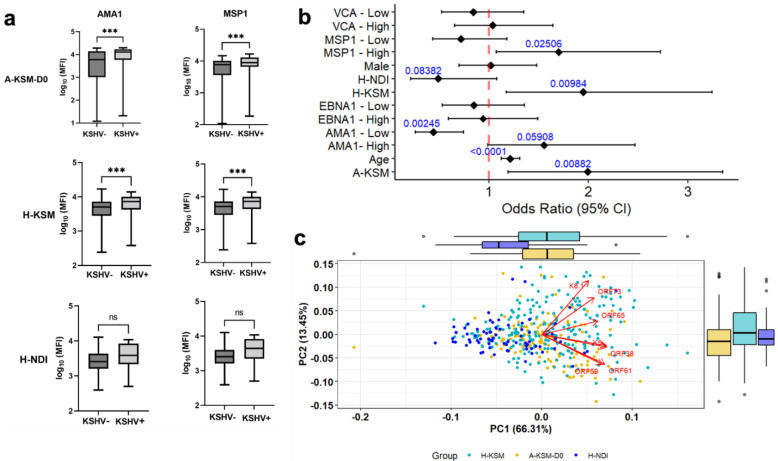
Distinct KSHV serological signature during acute malaria episodes. (**a**) Plot of anti-malarial IgG antibodies in KSHV seropositive relative to seronegative children from western Kenya. Significant differences are based on the Wilcoxon rank-sum test with Benjamini–Hochberg correction for multiple comparisons where appropriate (ns: *p* > 0.05; *** *p* < 0.001). Boxplots include the sample medians, minimum/maximum values, and first/third quartiles. (**b**) Forest plot of multivariate logistic regression analysis of determinants of KSHV seropositivity in the three study populations. (**c**) Study participants are clustered based on their anti-KSHV IgG antibodies into malaria-endemic vs. sporadic regions (PC 1) and acute malaria vs. healthy children from Kisumu (PC 2). Outer boxplots and whiskers show distribution of KSHV IgGs in A-KSM and H-KSM compared to H-NDI KSHV IgG combinations along both principal component 1 (PC1) and PC2. Abbreviations: A-KSM, children with acute malaria from Kisumu; H-KSM, healthy children from Kisumu; H-NDI, healthy children from Nandi; VCA, EBV Viral Capsid Antigen; EBNA1, Epstein–Barr nuclear antigen 1; MSP1, *P. falciparum* malaria Merozoite surface protein-1; AMA1, *P. falciparum* malaria Apical Membrane Antigen-1.

**Figure 2 ijms-24-06711-f002:**
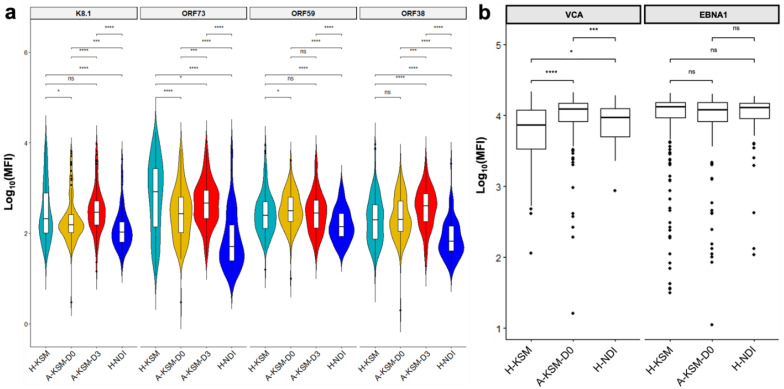
Increased IgG antibodies against KSHV lytic antigens during acute malaria episodes (**a**) Log-scale comparison of KSHV IgG antibodies against early (ORF59), late (K8.1) lytic, latency-associated (ORF73) antigens, and structural tegument protein ORF38 in A-KSM-D0 (cyan), H-KSM-D0 (yellow), H-KSM-D3 (red) and H-NDI (blue) (* *p* < 0.05; *** *p* < 0.001; **** *p* < 0.0001, ns: *p* > 0.05) (**b**) Log-scale comparison of EBV IgG antibodies against VCA during acute malaria infection. Significant differences are based on the Wilcoxon rank-sum test with Benjamini–Hochberg (BH) correction for multiple comparisons where appropriate (* *p* < 0.05; *** *p* < 0.001; **** *p* < 0.0001, ns: *p* > 0.05). Boxplots include the sample medians, minimum/maximum values and first/third quartiles. Abbreviations: KSHV, Kaposi sarcoma-associated herpesvirus; A-KSM-D0, acute Kisumu Day 0; A-KSM-D3, acute Kisumu Day 3; H-KSM, healthy Kisumu; H-NDI, healthy Nandi; VCA, Viral Capsid Antigen.

**Figure 3 ijms-24-06711-f003:**
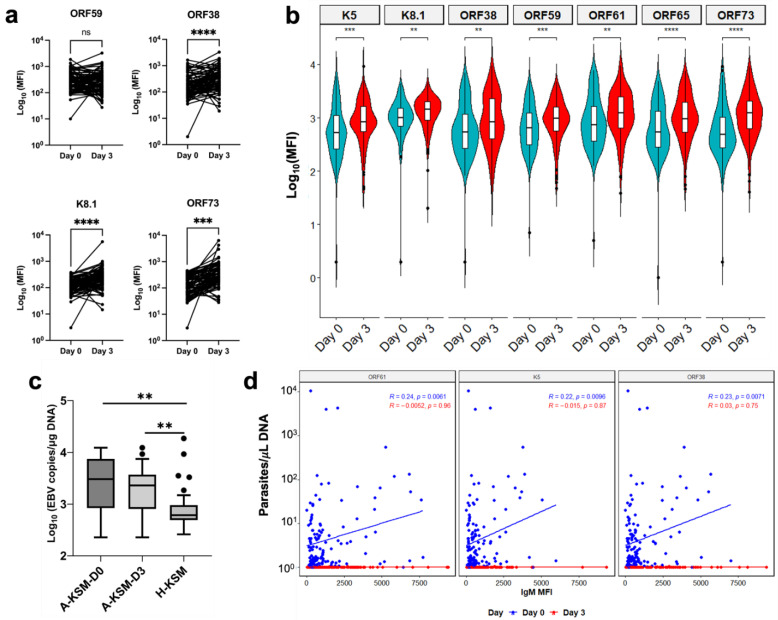
Variation in paired KSHV antibody responses during acute malaria episodes. (**a**) Paired comparisons of log-scale IgG antibodies against early (ORF59), late lytic (ORF38 and K8.1), and latency-associated antigen, ORF73 in KSHV-seropositive A-KSM children at Day 0 and 3 of malaria infection (Wilcoxon matched-pairs signed rank test; *** *p* < 0.001; **** *p* < 0.0001, ns: *p* > 0.05). (**b**) Paired comparison of log-scale IgM antibodies against 7 KSHV antigens at Days 0 and 3 of acute malaria infection (Wilcoxon matched pairs signed rank test with BH correction for multiple comparisons; ** *p* < 0.01; *** *p* < 0.001; **** *p* < 0.0001). (**c**) Comparison of EBV load in acute malaria-infected and healthy children from Kisumu (Wilcoxon rank-sum test, ** *p* < 0.01) Boxplots include the sample medians, minimum/maximum values, and first/third quartiles. (**d**) Spearman Correlation (r) test of parasite density and IgM antibodies against KSHV ORF61, K5 and ORF38 antigens. Abbreviations: KSHV, Kaposi sarcoma-associated herpesvirus; EBV, Epstein–Barr virus.

**Table 1 ijms-24-06711-t001:** Demographic characteristics of study participants.

	Malaria-Infected Children (A-KSM; *n* = 134)	Healthy Children-KSM (H-KSM; *n* = 221)	Healthy Children-Nandi (H-NDI; *n* = 77)	*p*-Value
Female, *n* (%)	63 (47%)	109 (49%)	38 (49%)	-
Age (years), Median (Range)	4.8 (1.7–9.9)	5.6 (1.3–15.8)	5.3 (0.5–16.6)	-
KSHV seropositive (*n*, %)	95 (71%)	171 (77%)	21 (28%)	<0.001 *

Abbreviations: KSHV, Kaposi sarcoma-associated herpesvirus; A-KSM-D0, acute Kisumu Day 0; A-KSM-D3, acute Kisumu Day 3; H-KSM, healthy Kisumu; H-NDI, healthy Nandi. * Statistical analysis performed by Pearson’s Chi-squared test for KSHV seropositivity count data.

## Data Availability

All data are provided in the manuscript and its Appendix A.

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
