# Peer review of "Distinctive Kaposi Sarcoma-Associated Herpesvirus Serological Profile during Acute Plasmodium falciparum Malaria Episodes"

_ijms, 2023, doi:10.3390/ijms24076711_

Round 1

Reviewer 1 Report

Kaposi Sarcoma and Malaria are dangerous diseases for people all over the world. In African countries, they kill a large number of people. The authors were in the centers of diseases and did a great job to study children from different places and the degree of morbidity. This study is very interesting for the scientists. I am sure this work will bring new knowledge in science for the prevention and treatment of these dangerous diseases.

Author Response

We would like to thank all the reviewers for their time and comments on our manuscript. Please find below response to comments that have led to revisions in our manuscript.

Reviewer 2 Report

The article entitled “Distinctive Kaposi Sarcoma-Associated Herpesvirus Serological Profile during Acute Plasmodium falciparum Malaria Episodes” discuss the seroprevalence of Kaposi sarcoma-associated herpesvirus among children in regions of high malaria endemicity in sub-Saharan Africa.

The article is well structured, comprehensive and the data are presented sound and clear.

I believe the article will be of interest for any medical professional involved in malaria surveillance and control.

Author Response

(The authors gave the same response as above.)

Reviewer 3 Report

Article written with great knowledge. Congratulations to the authors.

Author Response

We would like to thank you for their time and comments on our manuscript. 

Reviewer 4 Report

Concerns/Suggestions:

The lack of detectable KSHV in the acute malaria-infected cohort is significant and casts doubt on the idea of lytic reactivation during acute malaria.  Previous studies have detected KSHV in both blood and saliva suggesting that the virus is not anatomically sequestered. Suggest the authors soften language regarding lytic reactivation and offer any alternative explanations for an increase in seroreactivity.

The principal component analysis (Figure 1C) is not well described and as similar data are given in Figure 2A, 1C could be eliminated.

The authors state that IgG antibodies to both K8.1 and ORF73 are significantly higher in healthy subjects compared to Day 0 acute malaria subjects (Figure 2).  However, this is not evident for K8.1 as the difference is quite small and similar to the difference between ORF59 IgG in acute patients from Kisumu and healthy subjects from Kisumu, which is described as slightly elevated.

Figure 3A should be eliminated or the data presented in a different way as there are so many lines that most of the figure is a black box. It appears that for some individuals the IgG levels decreased from Day 0 to Day 3, for some it increased and for others stayed about the same. Perhaps the authors could comment.

In Figure 3B the overall trend is for an increase in IgM levels across the time period. Given that some of the antigens are associated with latency, some with the early lytic and some with late lytic stage, can the authors explain the similar trend for all antigens?

 Most of the text in the figures is too small to be readable. This is particularly noticeable in the top of Figure 2A but is a problem in general. 

The text refers to supplementary figures S1A, S1B, S1C and S1D.  However, the Supplementary Materials section refers to S1, S2, S3 and Table S1, which is what was provided.   The description of S2 in Supplementary Materials does not match the figure labelled a S2.  Similarly, the text states’…. correlation with IgG against KSHV antigens, including K5, ORF38, ORF59 and ORF61 (Figure S1B) but Figures S2 gives IgM results for K8.1, ORF59, ORF65 and ORF73.  It is unclear what Figures S1B refers to.

Minor concerns/Typos:

The x axis of the lower panel in Figure 1A is missing or cut off.  The abbreviations VCA and EBNA-1 are used in Figure 1B but not discussed in the text until Figure 2 is described.  Abbreviations AMA-1, MSP1, VCA and EBNA1 are not included in the figure legend.

Reviewer 5 Report

In this clinical study, the authors reported that KSHV seroprevalence was significantly associated with malaria endemicity in pediatric patients. They also found that KSHV serological profiles during acute malaria episodes were distinct from age-matched non-malaria-infected children, indicating that acute malaria episodes led to perturbation of KSHV latency in seropositive children. Although this is an interesting study, one major issue is that some of their data seem not fully support their conclusions (see below my comments):

1. In Figure 1A, the group labels in H-NDI are missing.

2. In Figure 2A, they mentioned “Anti-ORF73 and K8.1 IgGs were significantly higher in healthy subjects compared to acute malaria subjects from Kisumu at day 0”. Can they explain why the results look like that? Especially, K8.1 is a lytic protein, why its IgG level is NOT higher in acute malaria subjects at day 0 if they think “acute malaria episodes led to perturbation of KSHV latency in seropositive children”?

3. In Figure 3, can they explain why some lytic gene IgG level (ORF59) is higher in A-KSM at day 0, and the other lytic gene IgG level (ORF38, K8.1) is higher in A-KSM at day 3? My understanding is anti- malaria treatments will cause the IgGs levels of all lytic genes decreased, if their conclusion is correct, right?

4. If “acute malaria episodes led to perturbation of KSHV latency”, why in the acute malaria-infected cohort, KSHV load was undetectable both in whole cell pellets and cell-free plasma?? (not like EBV situations shown in Figure 3C)? And did they examine KSHV load in A-KSM at day 3, too??

5. Are their findings in pediatric patients similar or different to those reported in adult patients with malaria, which should be discussed at least.

6. What status of HIV infection in these African children, does HIV affect Pf infection?

Round 2

Reviewer 5 Report

The authors have addressed correctly all of my comments.